# Serum Lipidome Signatures of Dogs with Different Endocrinopathies Associated with Hyperlipidemia

**DOI:** 10.3390/metabo12040306

**Published:** 2022-03-30

**Authors:** Nadja S. Sieber-Ruckstuhl, Wai Kin Tham, Franziska Baumgartner, Jeremy John Selva, Markus R. Wenk, Bo Burla, Felicitas S. Boretti

**Affiliations:** 1Clinic for Small Animal Internal Medicine, Vetsuisse Faculty, University of Zurich, 8057 Zürich, Switzerland; franziska.baumgartner@hotmail.com (F.B.); fboretti@vetclinics.uzh.ch (F.S.B.); 2Precision Medicine Translational Research Program and Department of Biochemistry, Yong Loo Lin School of Medicine, National University of Singapore, 8 Medical Drive, Singapore 117597, Singapore; e0681551@u.nus.edu (W.K.T.); bchjjs@nus.edu.sg (J.J.S.); bchmrw@nus.edu.sg (M.R.W.); 3Agilent Technologies Singapore Pte. Ltd., 1 Yishun Ave 7, Singapore 768923, Singapore; 4Singapore Lipidomics Incubator, Life Sciences Institute, 28 Medical Drive, National University of Singapore, Singapore 117456, Singapore

**Keywords:** thyroid, dyslipidemia, hypercortisolism, atherosclerosis, metabolic syndrome, lipidomics, mass spectrometry

## Abstract

Hyperlipidemia (hypertriglyceridemia, hypercholesterolemia) is a common finding in human and veterinary patients with endocrinopathies (e.g., hypothyroidism and hypercortisolism (Cushing’s syndrome; CS)). Despite emerging use of lipidomics technology in medicine, the lipid profiles of these endocrinopathies have not been evaluated and characterized in dogs. The aim of this study was to compare the serum lipidomes of dogs with naturally occurring CS or hypothyroidism with those of healthy dogs. Serum samples from 39 dogs with CS, 45 dogs with hypothyroidism, and 10 healthy beagle dogs were analyzed using a targeted lipidomics approach with liquid chromatography-mass spectrometry. There were significant differences between the lipidomes of dogs with CS, hypothyroidism, and the healthy dogs. The most significant changes were found in the lysophosphatidylcholines, lysophosphatidylethanolamines, lysophosphatidylinositols, phosphatidylcholines, phosphatidylethanolamines, phosphatidylglycerols, ceramides, and sphingosine 1-phosphates. Lipid alterations were especially pronounced in dogs with hypothyroidism. Several changes suggested a more atherogenic lipid profile in dogs with HT than in dogs with CS. In this study, we found so far unknown effects of naturally occurring hypothyroidism and CS on lipid metabolism in dogs. Our findings provide starting points to further examine differences in occurrence of atherosclerotic lesion formation between the two diseases.

## 1. Introduction

Hyperlipidemia is an increased concentration of lipids in the blood; it includes increases in triglycerides, cholesterol, or both [1,2]. Secondary hyperlipidemia is the most common form of hyperlipidemia in veterinary medicine; the most important causes of secondary hyperlipidemia are endocrine diseases (e.g., hypothyroidism and hypercortisolism (Cushing’s syndrome; CS)) [1,2]. CS is one of the most common endocrinopathies in dogs [3]. In CS, important changes in lipid metabolism induced by chronic increased cortisol concentrations include enhanced lipolysis of subcutaneous adipose tissue, accompanied by visceral adiposity. During disease progression, dogs develop disorders similar to human metabolic syndrome; hyperlipidemia, hypertension, and secondary diabetes mellitus are frequently observed. Based on this similarity with human metabolic syndrome, dogs are used as a model to study hyperlipidemia, obesity, and diabetes mellitus [4].

Thyroid hormones also affect lipid metabolism [1,2]. When serum thyroid hormone concentrations are low (hypothyroidism), lipid synthesis and degradation are depressed. A net accumulation of lipids results because the effect on degradation is greater than the effect on synthesis. The increases in cholesterol and triglycerides are similar to those in patients with CS. Atherosclerotic lesions have been described in dogs with hypothyroidism; these changes are absent in dogs with CS [5]. This difference does not occur in humans. Both endocrinopathies are associated with atherosclerosis and negative effects on cardiovascular health. The reason for this phenomenon in dogs is unknown. Although atherosclerosis in itself is a rare disease in dogs and lipid deposition is different compared with humans, dogs are the only domestic animal known to have measurable amounts of lipid deposition (e.g., cholesterol) in the arteries [6].

Lipidomics defines and quantifies the lipid species profiles within a biological system [7]. This powerful tool is used to assess possible pathomechanisms and biomarkers of disease. Measurement of the classical plasma lipids (i.e., triglyceride, cholesterol, high-density lipoproteins (HDLs), and low-density lipoproteins (LDLs)) might be replaced by determination of new biomarkers discovered in lipidomics studies evaluating the blood lipidome in pathological states [8,9]. Novel scores to predict cardiometabolic risk in humans based on plasma concentrations of specific ceramides and other lipid species are being developed [10,11].

Changes in the lipidome have been well-characterized in an experimental dog model of chronic hypercortisolemia [12]. The lipids that were especially affected were monohexosylceramides, phosphatidylinositols, ether phosphatidylcholines, acyl phosphatidylcholines, triacylglycerols, and sphingosine 1-phosphates. However, there have been no lipidomic studies of dogs with naturally occurring hypercortisolemia. Moreover, the lipidome of dogs with hypothyroidism has not been evaluated, either experimentally or in dogs with naturally occurring diseases. Such studies, however, would be of great interest because both endocrinopathies are associated with severe hyperlipidemia.

The objectives of this study were to characterize and compare serum lipidomes of dogs with naturally occurring CS and those of dogs with hypothyroidism using a targeted liquid chromatography-tandem mass spectrometry (LS-MS) approach. Lipid profiles of healthy beagle dogs were compared to those of dogs with CS or hypothyroidism.

## 2. Results

### 2.1. Study Population

Ten healthy beagles (HBs), forty-five dogs with hypothyroidism (HT), and thirty-nine dogs with Cushing’s syndrome (CS) were included in the study (Table 1).

### 2.2. Overall Lipidomic Profiles

The targeted lipidomics analysis resulted in a dataset with 301 quantified lipid species that were in the phospholipid, sphingolipid, glycerolipid, or cholesterol ester lipid classes (Appendix A). Principal component analysis (PCA) revealed partial separation of the HB, HT, and CS groups (Figure 1). The loading analysis of the PCA revealed a group of lipid species that include lysophosphatidylcholine (LPC), lysophosphatidylserine (LPS), phosphatidylcholine (PC), phosphatidylethanolamine (PE), sphingomyelin (SM), and ceramide (Cer) species in the direction of principal component 1 that partially separated HT from CS and HB. Furthermore, another group comprising diacylglyceride (DG) and triglyceride (TG) species was present in the direction of the principal component 2 dimension that contributed to the partial separation of HB from HT and CS. The lipidomics as well as the clinical chemistry measurements (Appendix A) revealed that the two outliers in the CS group had very high total TG concentrations.

### 2.3. Serum Cholesterol and Triacylglycerides

Serum TG concentrations were determined using colorimetric assay and MS-based lipidomics assays. TG concentrations determined using both assays were significantly higher in the HT and CS groups, compared with the HB group (Figure 2A). TG concentrations were also significantly higher in the HT compared with the CS group. DGs were higher in the HT, compared with the HB and CS groups.

Esterified cholesterol (cholesteryl esters, CEs) was measured using the MS-based assay, summing up all quantified CE species (Figure 2B). CEs were also increased in the HT compared with the CS groups. Compared with the HB-group dogs, CEs exhibited a tendency for increased concentrations in the HT-, but not in the CS-, group dogs. Serum total cholesterol (TC) was measured using a colorimetric assay. TC was significantly elevated in dogs with HT, compared with dogs with CS. Compared with the HB group, concentrations were increased in both the HT and CS groups (Figure 2B).

The ratio between esterified and total cholesterol was significantly decreased in the HT and CS groups, compared with the HB group; the concentrations in the HT group were significantly lower than in the CS group (Figure 2C).

### 2.4. Diacyl Phospholipids

All measured diacyl phospholipid classes (phosphatidylcholine (PC), phosphatidylethanolamine (PE), phosphatidylglycerol (PG), and phosphatidylinositol (PI)) were significantly increased in the HT compared with the HB group, and in the CS compared with the HB group, except PI (Figure 3). Concentrations of all these classes were significantly higher in the HT compared to the CS group.

### 2.5. Ether and Plasmalogen Phospholipids

The CS group had significantly lower concentrations of ether (alkyl) and plasmalogen (alkenyl) PC (PC-O and PC-P, respectively) than the HB group (Figure 4). The HT group had lower concentrations of PC-O, compared with the HB group; and PC-P concentrations in the HT group were higher than in the CS group (Figure 4). There were no significant between-group differences in ether or plasmalogen PE (PE-O and PE-P, respectively) (Figure 4).

### 2.6. Lysophospholipids

There were significant differences in all measured lysophospholipid classes (Figure 5). In the HT group, concentrations of lysophosphatidylcholine (LPC), ether lysophosphatidylcholine (LPC-O), plasmalogen lysophosphatidylcholine (LPC-P), lysophosphatidylethanolamine (LPE), plasmalogen lysophosphatidylethanolamine (LPE-P), and lysophosphatidylserine (LPS) were higher than in the CS and HB groups. In the CS group, concentrations of LPC and LPC-P were lower, and concentrations of LPE were higher, than in the HB group. Lysophosphatidylinositol (LPI) concentrations were significantly lower in the HT than the HB group; they were significantly lower in the CS than in the HT group.

### 2.7. Sphingolipids

We found significantly increased concentrations in the HT compared with the HB and CS groups for all analyzed ceramide classes (Cer, containing d18:0, d18:1, or d18:2 sphingoid bases, Figure 6A), and for three of the four individual ceramide species Cer d18:1/16:0, Cer d18:1/18:0, Cer d18:1/24:0, and Cer d18:1/24:0 (Figure 6B). Among the glycosphingolipids, hexosylceramides (Hex1Cer) and gangliosides (GM3) were also increased in the HT group; there were no significant differences for the di- and trihexosylceramides (Hex2Cer and Hex3Cer) (Figure 6C). Sphingomyelins (SMs) were also increased in the HT-group dogs (Figure 6D). Sphingosine 1-phosphate (S1P) was significantly increased in the HT compared with the HB group; S1P was even more increased in the CS group (Figure 6E)

Ceramide ratios (Figure 7) were significantly lower in the HT group for [Cer d18:1/16:0]/[Cer d18:1/24:0] and [Cer d18:1/18:0]/[Cer d18:1/24:0], compared with the CS and HB groups. The [Cer d18:1/18:0]/[Cer d18:1/24:0] ratio was increased in the CS group compared with the HB group.

## 3. Discussion

This study focused on quantification of a large panel of lipids in the blood of dogs with naturally occurring HT and CS, which are two endocrinopathies frequently associated with hyperlipidemia [1]. Samples from 10 healthy beagle dogs were analyzed for comparison.

In both diseases, increases in TC, increases in TG, or increases in both parameters can be seen, although some studies report hypercholesterolemia to be more common than hypertriglyceridemia [13]. The magnitudes of TC and TG elevation can range from mild to marked [1]. In dogs with CS, hyperlipidemia can result from downregulation of LDL receptors and decreased liver uptake of LDL and/or development of insulin resistance, which impairs lipoprotein lipase activity [14,15]. In HT, the specific mechanism for development of hyperlipidemia remains unknown. However, a thyroxine-deficiency-associated decrease in lipoprotein lipase activity is likely [16]. Decreases in hepatic LDL receptor activity and in hepatic lipase are also proposed mechanisms for the TC abnormalities observed in dogs with HT [17].

In this study, dogs with HT had significantly higher TC, CE, and TG concentrations than dogs with CS. Why the hyperlipidemia was more severe in the dogs with HT is unclear. Neither of the two diseases is reported to have consistently more severe lipid changes than the other. A study including dogs with severe hypertriglyceridemia (>3.39 mmoL/L) found that 43.5% suffered from CS, 40.3% from HT, and 8.1% from a combination of HT and CS [18]. One possible reason for more severely increased TC and TG concentrations in dogs with HT is the presence of a more pronounced and long-standing disease. Signs of HT (e.g., lethargy, body weight increase) are usually mild and only slowly progressive. In contrast, signs of CS (e.g., polyuria, polydipsia, panting) are often very obvious and these signs lead owners to rapidly seek veterinary assistance, often during an early phase of the disease.

### 3.1. Cholesterol Metabolism

Dogs with HT had relatively lower ratios of esterified cholesterol to total cholesterol (Figure 2C), compared with HB- and CS-group dogs. This result suggested the presence of higher proportions of free versus esterified cholesterol in the dogs with HT. A CE/TC ratio >1, as found for the HBs (Figure 2C), can be explained by errors in absolute quantification that led to overestimation of cholesteryl esters in the lipidomics assay. Concentrations reported by MS-based lipidomics approaches are not considered as absolute quantifications [19,20]. However, the CE concentrations and thus also the CE/TC ratios are likely comparable between groups.

Decreased lecithin–cholesterol acyltransferase (LCAT) activity can occur in rats with induced hypothyroidism [21,22]. However, a study of humans revealed that the hypothyroid state has no effect on LCAT concentrations [23]. Therefore, most of the increase in unesterified cholesterol in the serum in the hypothyroid state seemed to be associated with the LDL cholesterol increase [23].

### 3.2. Phospholipids Increased in Both Conditions, HT and CS

PCs and PEs are very abundant glycerophospholipids in mammals. PCs are structural components of biological membranes. They are involved in gene regulation and homeostatic control of serum glucose concentrations [24]. PEs are essential for cell division, mitochondrial respiratory function, and membrane topology and function [25,26]. In the HT-group dogs, all measured diacyl phospholipid classes were significantly higher than in the other two groups. Thyroid hormones are known to affect lipid composition. Our results were consistent with study findings in humans; PC and PE concentrations are increased in patients with hypothyroidism and in pregnant women with subclinical hypothyroidism [27,28]. In the CS-group dogs, the PC, PE, and PG concentrations were significantly increased, compared with the HB dogs. However, CS-group dogs had significantly lower PC, PE, and PG concentrations than the HT-group dogs. This result was inconsistent with the results of Di Dalmazi and coworkers. They found that human patients with CS have lower concentrations of some phosphatidylcholines than control patients [29]. There were significant differences in the phospholipid classes between the HT versus CS-group dogs. Interestingly, phosphatidylcholine concentrations in blood or ingested food seem to be associated with atherosclerosis in humans [30,31]. Atherosclerosis is a rare disease in dogs. Most often, atherosclerosis is associated with endocrine disease that leads to hyperlipidemia, particularly HT [5,32]. Why dogs with HT seem more prone to develop atherosclerosis than dogs with other endocrine diseases (e.g., CS) is not clear. The increased PC and PE concentrations combined with other herein described lipid alterations might be important contributing factors.

### 3.3. Lysophospholipids Distinguish HT from CS

All lysophospholipid classes were significantly increased in the HT group. However, the effect seemed especially pronounced in LPCs. In the CS group, the LPCs were significantly decreased. Study results suggest that LPCs are important mediators of atherosclerosis [33]. High concentrations of LPC induce overproduction of nitric oxide, which leads to increased oxidative stress and endothelial cell injury [34]. The differences between the HT and CS groups found in this study suggested a more atherogenic serum profile in dogs with HT.

### 3.4. Ceramides and Sphingosine 1-Phosphates

In the dogs with CS, the SM and most of the ceramides were not different, compared with the healthy dogs. This result was consistent with the results of a study in human patients with CS [29]. The authors found no alterations in SM concentrations in serum; they speculated that the finding indicates that the plasma membrane composition and activities of the SM cycle and ceramide concentrations are not affected by a hypercortisolemic state [29]. In contrast, the HT-group dogs had significantly higher ceramide concentrations than the healthy dogs or dogs with CS. Ceramides are a family of lipid molecules, which are transported by lipoproteins in the blood [35]. Ceramides are associated with many pathological states, and study results suggest that circulating ceramides have an important role in the pathogenesis of cardiovascular disease in humans [35,36,37,38]. In humans, specific ceramides seem to be associated with development of atherosclerotic vascular disease and its progression to adverse events and death [38,39]. Higher concentrations of three ceramides, Cer(d18:1/16:0), Cer(d18:1/18:0), and Cer(d18:1/24:1), and higher ratios of these molecules with Cer(d18:1/24:0), seem to especially increase this risk [38,39]. The results for a fourth ceramide, Cer(d18:1/24:0), have been inconsistent. One study found a lower risk with increased concentrations of Cer(d18:1/24:0); other studies found positive correlations with the incidence of cardiovascular events [38,40]. The three ceramides, Cer(d18:1/16:0), Cer(d18:1/18:0), and Cer(d18:1/24:1), and their ratios to Cer(d18:1/24:0), are used to calculate scores (e.g., CERT1) used for risk stratification in the clinical setting [35].

This study revealed significant alterations in three of four of the above-mentioned ceramides in dogs with HT. Dogs with HT had significantly higher concentrations of Cer(d18:1/16:0), Cer(d18:1/24:0), and Cer(d18:1/24:1) than dogs with CS or HB. This study was the first to find increased ceramide concentrations in dogs with HT. Because these three ceramides seem to be important in the pathogenesis of cardiovascular disease, and dogs with HT seem more prone to develop atherosclerosis than dogs with CS, these findings seem to confirm a connection between ceramides and cardiovascular risk and suggest a more atherogenic serum profile could be present in dogs with HT. The ratios of Cer(d18:1/16:0) and Cer(d18:1/24:1) to Cer(d18:1/24:0) were significantly lower in the HT group than in the CS- or HB-group dogs. As mentioned above, the role of Cer(d18:1/24:0) during development of cardiovascular disease remains unclear. Further studies are needed to clarify the roles of Cer(d18:1/24:0) in the development of cardiovascular problems in general, and particularly in dogs.

Other classes of sphingolipids (e.g., HexCer) can also affect cardiovascular death risk in humans [11]. The measured HexCer and Hex2Cer concentrations were not significantly different between the two disease groups in our study. However, in both groups, HexCer concentrations were significantly higher than in the HB-group dogs. The increased HexCer concentrations could be attributed to an increase in the precursor ceramide, which seems especially true for dogs with HT. Increased HexCer concentrations, however, can also occur as a result of an altered catabolism of complex glycosphingolipids. A study of dogs with experimentally induced Cushing’s syndrome found increased HexCer concentrations [12]. Again, as in the earlier study, no significant changes in Hex2Cer concentrations were found in this study [12].

Study findings for the relationships between sphingolipids and increased risk for cardiovascular events in humans are inconsistent. However, the large differences found for several sphingolipids in this study between HT and CS suggest a higher risk for cardiovascular complications in dogs with HT and may highlight the importance of further research in the sphingolipid field in humans and animals. S1P was significantly increased in both disease states in this study, but the increase was more severe in the dogs with CS than in the dogs with HT. S1P is a potent lipid mediator with a variety of physiologic effects and seems also to be involved in diseases such as atherosclerosis, cancer, and diabetes mellitus [41]. The effects of S1P in atherosclerosis remain unclear (anti- and pro-atherosclerotic properties) [41]. Most likely, specific effects are associated with the unique characteristic distribution of S1P in the circulation. In the circulation, S1P is carried on HDL (about two-thirds) or by albumin [42]. The S1P bound to HDL has been found to have mainly beneficial effects, such as nitric-oxide-mediated vasodilation, anti-oxidative, anti-apoptotic, and anti-inflammatory effects, and direct cardioprotection against reperfusion injury [41,43,44]. Whether S1P in the circulation of dogs has the same characteristic distribution as in humans is unknown. It is also unknown if there are differences in this distribution between the two diseases. This area of study is of great interest considering the atherogenic differences between the two diseases.

This study had some limitations. First, the question of biological variation. The HT- and CS-group dogs were client-owned and had various dietary habits and geographical differences in origin. Further, the client-owned dogs had different breeds, sexes, and neutering status. The HB-group dogs were from the same breed (Beagle dogs), lived in the same environment, and consumed the same diet, but also were of different sexes and neutering status. These differences may have affected our results, as no standardization was possible. Second, the cardiovascular system was only assessed based on a routine clinical examination, including heart rate, rhythm and murmurs and pulse rate, rhythm, and strength. A more specific examination of the cardiovascular system including, e.g., echocardiography or angio-computer tomography is only performed if a specific suspicion arises during routine examination (heart murmur, suspicion of thromboembolism). Therefore, no statement about the in-depth cardiovascular health (e.g., presence or absence of atherosclerosis) of the included dogs is possible.

## 4. Materials and Methods

### 4.1. Animals and Sample Collection

Samples from 10 healthy purpose-bred Beagle dogs were used for this study (3 females, 7 males). The median (range) age was 6 years (2–7 years) and the median (range) body weight was 14.8 kg (10.0–18.9 kg). They were considered healthy based on the results of physical examinations, biochemical investigations of serum and urine, and complete blood counts. The Beagle dogs received dry adult maintenance dog food for at least one month before the study (Josera, Adult Sensitive, Kleinheubach, Germany), had free water access, and were housed in small groups of 2–4 dogs per kennel at the research center of the Vetsuisse Faculty of the University of Zurich. After an overnight fast (≥12 h), blood samples were collected into serum containers (Vacuette Greiner Bio-One, Frickenhausen, Germany). The study was approved by the Cantonal Veterinary Office of Zurich (TVB 276/2014) and conducted in accordance with guidelines established by the Animal Welfare Act of Switzerland.

Samples from 45 client-owned dogs with naturally occurring hypothyroidism were included in the study. The diagnosis was based on serum total thyroxine (TT4) concentrations < 0.7 μg/dL and a concurrent serum thyroid-stimulating hormone (TSH) concentration > 83 mU/L (Table 2). Cut-off values for inclusion were chosen substantially below and above the corresponding reference ranges for hypothyroidism, to confirm the diagnosis [45]. All samples were from Idexx laboratory (Idexx GmbH, Germany). Serum T4 concentrations were determined using DRI Thyroxine (T4) Assay (Microgenics Corporation (part of Thermo Fisher Scientific), Freemont, CA, USA) using an Olympus AU5800, Beckman Coulter Analyzer. The serum TSH concentration was determined using a solid part, 2-site chemiluminescent enzyme immunometric assay, Siemens, Immulite 2000/Xpi. The reference interval used for T4 was 1.0–4.0 µg/dL; it was <29 mU/L for TSH. The median (range) age of the dogs was 6 years (3–14 years); the age of three dogs was unknown. Body weights of the dogs were unknown. Of the 45 dogs, 23 were females, 18 were males; the sex of the remaining 4 dogs and the neutering status of all dogs were unknown.

Samples from 39 dogs with naturally occurring Cushing’s syndrome were also included in the study. All these dogs were client-owned and were presented to the Clinic for Small Animal Internal Medicine Zurich at the University of Zurich in Switzerland between 2010 and 2016. Diagnosis was based on a positive low-dose dexamethasone suppression test (LDDS test) and/or a positive ACTH-stimulation test (Table 2). Dogs with concurrent disease (e.g., diabetes mellitus, chronic renal disease) were excluded from the study. The median (range) age was 11 years (5–16 years) and the median (range) body weight was 14.3 kg (2.1–42.9 kg). Of the 39 dogs with Cushing’s syndrome, 21 were males (13 castrated) and 18 were females (15 spayed). For the ACTH-stimulation test, 5 μg/kg synthetic tetracosactide (Synacthen^®^, Novartis Pharma Schweiz AG, Bern, Switzerland, Pharmacode 6748610) was injected intravenously and serum cortisol concentrations were measured at 0 and 1 h after injection. For the low-dose dexamethasone suppression test, 0.01 mg/kg dexamethasone (Dexadreson^®^, MSD Animal Health GmbH, Luzern, Switzerland, ATCvet number: QH02AB02) was injected intravenously and serum cortisol concentrations were measured 0, 4, and 8 h after injection. Serum cortisol concentrations were measured using a competitive immunoassay validated for dogs (DPC Immulite 2000, Siemens Schweiz AG, Zurich, Switzerland). The intra-assay coefficients of variation were 10.0% and 6.3% at cortisol concentrations of 2.7 and 18.9 µg/dL, respectively. The assay sensitivity was 0.2 µg/dL [46].

All blood samples were centrifuged after clot formation. The serum was removed and stored at −80 °C until analysis.

All clinical blood serum chemistry parameters were determined at the Clinical Laboratory, Vetsuisse Faculty, University of Zurich, using a Cobas Integra 800 instrument (Roche Diagnostics AG, Rotkreuz, Switzerland).

### 4.2. Lipid Extraction and Derivatization

Lipid extraction and derivatization were performed as previously described [47]. Briefly, lipids were extracted using a butanol:methanol single-phase, single-step liquid-liquid extraction method [48]; 10 μL aliquots of the serum samples were each mixed with 90 μL of extraction mix consisting of 1-butanol:methanol (1:1, *v*/*v*) and internal standards (ISTD; Appendix A). After sonication for 15 min in a water bath kept at 20 °C, the samples were centrifuged at 14,000× *g* for 10 min at 4 °C and the supernatant was transferred to autosampler vials for subsequent LC-MS analysis. Aliquots of batch quality control (BQC) samples, generated by pooling equal volumes of each serum sample before extraction, were co-extracted and analyzed with study samples at fixed intervals. Process blank samples, which did not contain any serum, were also co-extracted and analyzed in the same way as the samples. For the measurement of sphingosine 1-phosphate (S1P) [49], 50 μL aliquots of the lipid extracts were diluted with 50 μL methanol and derivatized with 20 μL trimethylsilyldiazomethane (2 mol/L in hexanes, Acros Organics, Thermo Fisher Scientific, Freemont, CA, USA) for 20 min at 25 °C and 700 rpm (Thermomixer, Eppendorf, Germany). The derivatization reaction was stopped using 1 μL glacial acetic acid. Derivates were then centrifuged at 14,000× *g* for 10 min at 4 °C, and supernatants were transferred into autosampler vials for subsequent liquid chromatography-mass spectrometry (LC-MS) analysis. Response QC (RQC) samples were prepared by serial dilutions of pooled BQC extracts or derivates, respectively, with 1-butanol:methanol (1:1, *v*/*v*).

### 4.3. LC-MS Analyses

Lipids were analyzed using targeted LC-MS as previously described [12,47]. The LC-MS system consisted of an Agilent 1290 infinity ultra-high-performance liquid chromatography pump and an Agilent 6495A triple quadrupole (QQQ) mass spectrometer (Appendix A). All analyses were performed in dynamic multiple reaction monitoring mode with unit resolution. Monitored transitions are presented in Appendix A. Peak integration was performed using Agilent MassHunter Quantitative Analysis software (Version B.08) [12]. Isotopic correction of peak areas was performed for lipid species that were co-integrated with M+1 or M+2 isotopes of other species due to lack of chromatographic separation [50]. Drift correction was performed for lipid species where the correction led to a reduction in the coefficient of variation (CV) of the study samples by >2%. This drift correction was based on the BQC samples and used loess smoothing (span 0.75) on log2-transformed peak areas [51].

The concentration of lipid species was calculated based on the corresponding class-specific ISTD (Appendix A). For lipid classes where normalization led to an increase in the CV of study samples, normalization and quantification were based on the corresponding average ISTD peak area in the BQC samples. Quality control filtering criteria for a lipid species to be considered for further analysis were: (i) signal-to-blank ratio > 5 (median peak area of BQC compared to Process Blank samples), (ii) a raw peak area above 500, (iii) analytical CV < 25% (based on normalized peak areas in BQCs), and (iv) linear response of the RQC dilution series R^2^ > 0.8. Calculated lipid species concentrations, based on the corresponding ISTD, were expressed as μmol/L serum (Appendix A). Total concentrations of lipid classes were obtained by summing up the concentration of all species of a corresponding class. All data processing and QC filtering were performed in R (Version 4.1.0) [52].

### 4.4. Statistics and Visualization

R (Version 4.1.0) was used for statistics and visualizations [52]. The principal component analysis (PCA) was generated using the R package PCAtools [53] from scaled, centered log2-transformed lipid abundance values. Box/dot plots were generated using the R packages ggplot (Version 3.3.5) [54] and ggsignif (Version 0.6.3) [55] with the help of additional tidyverse packages [56].

## 5. Conclusions

This study is the first serum lipidomic study of dogs with naturally occurring Cushing’s syndrome and dogs with hypothyroidism. Our results revealed widespread and often different changes in the serum lipidomes of dogs with HT and CS. Many alterations were more pronounced in dogs with HT. The most important changes were found in the following lipid classes: LPC was the only lipid class with distinct changes in opposing directions in HT and CS compared with HB. It increased in dogs with HT, but decreased in dogs with CS, which suggested it was involved in the pathogenesis of concurrent diseases. Ceramide lipid species, identified as biomarkers for cardiovascular diseases in humans, increased only in HT and remained unchanged in dogs with CS. Taken together, the findings suggest the presence of a more atherogenic serum profile in dogs with HT than in dogs with CS. Future studies should further evaluate and compare lipid species between the two diseases and before and after specific interventions to provide deeper insights into this interesting topic.

## Figures and Tables

**Figure 1 metabolites-12-00306-f001:**
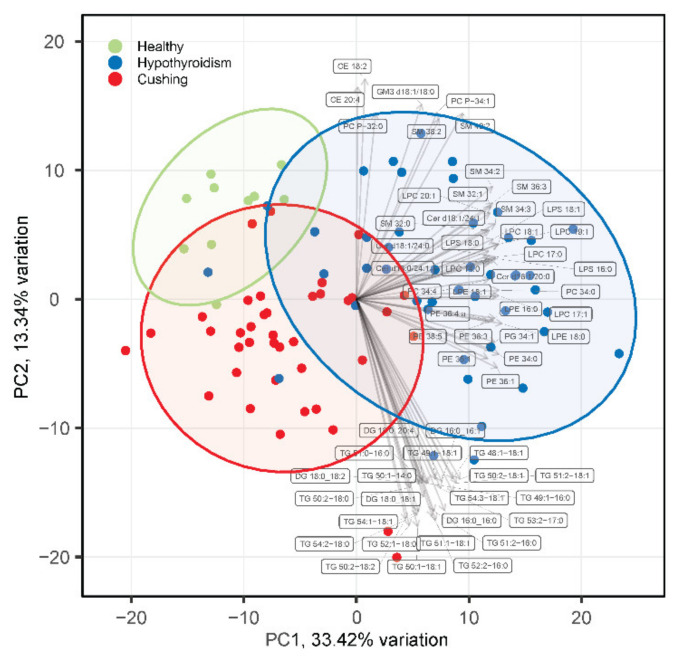
PCA analysis of serum lipidomes of healthy beagles (green), dogs with hypothyroidism (blue), and dogs with CS (red). The top 30 loading vectors are overlaid with corresponding lipid species names indicated. PC1 and PC2 in the axis labels correspond to principal component 1 and 2, respectively.

**Figure 2 metabolites-12-00306-f002:**
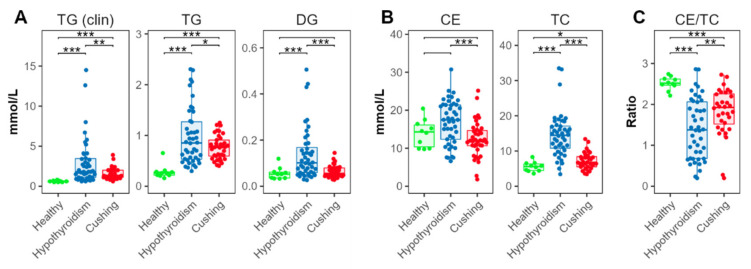
Triglyceride and cholesterol concentrations. (**A**) Total triglyceride (TG) and diacylglyceride (DG). (**B**) Total esterified (cholesteryl esters, CE) and total cholesterol (TC) concentrations. (**C**) Ratios between esterified and total cholesterol. TG (clin) and TC were determined using colorimetric clinical chemistry assays, the other lipid classes via the lipidomics platform. Levels of statistical significance between two groups calculated by two-tailed Welch’s *t*-tests are indicated with stars: * *p* ≤ 0.05, ** *p* ≤ 0.01, *** *p* ≤ 0.001.

**Figure 3 metabolites-12-00306-f003:**
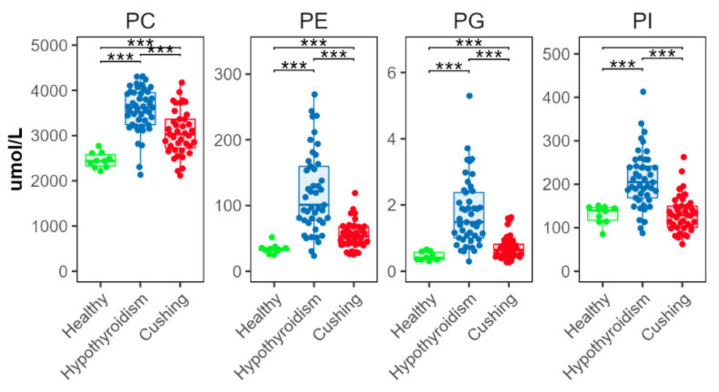
Total abundance of species from major diacyl phospholipid classes. Levels of statistical significance between two groups calculated by two-tailed Welch’s *t*-tests are indicated with stars: *** *p* ≤ 0.001.

**Figure 4 metabolites-12-00306-f004:**
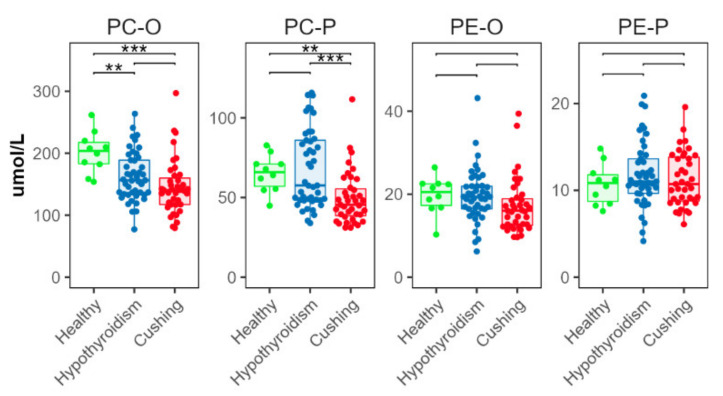
Total abundance of species from ether (alkyl, -O) and plasmalogen (alkenyl, -P) phosphatidylcholine (PC) and phosphatidylethanolamine (PE) classes. Levels of statistical significance between two groups calculated by two-tailed Welch’s *t*-tests are indicated with stars: ** *p* ≤ 0.01 and *p* > 0.001, *** *p* ≤ 0.001.

**Figure 5 metabolites-12-00306-f005:**
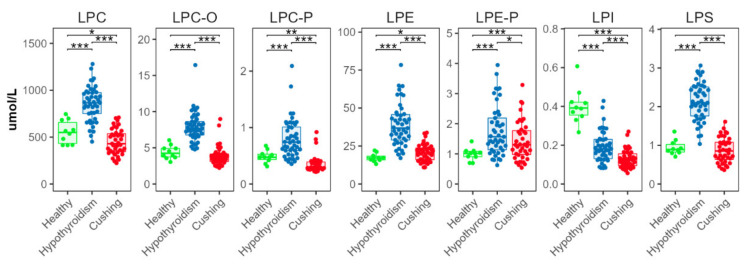
Total abundances of species from lysophospholipid classes. Levels of statistical significance between two groups calculated by two-tailed Welch’s *t*-tests are indicated with stars: * *p* ≤ 0.05, ** *p* ≤ 0.01, *** *p* ≤ 0.001.

**Figure 6 metabolites-12-00306-f006:**
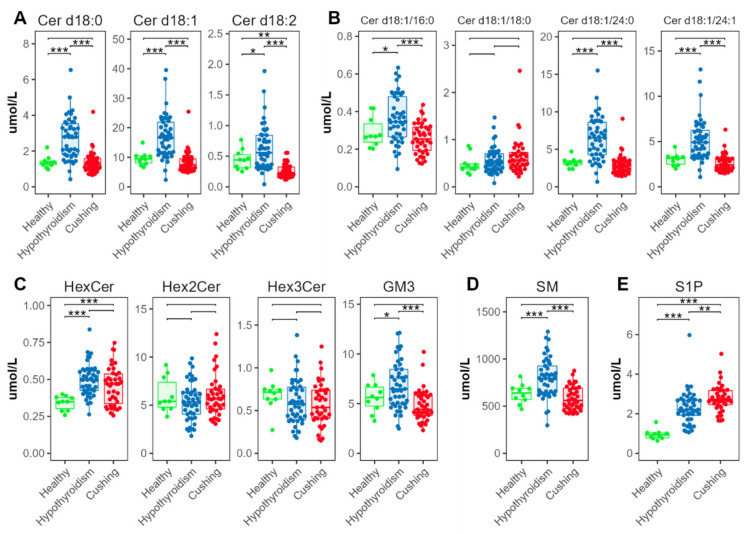
Abundance of species from sphingolipid classes. (**A**) Total ceramide, (**B**) individual ceramide species, (**C**) glycosphingolipids, (**D**) sphingomyelins, (**E**) sphingosine 1-phosphates. Levels of statistical significance between two groups calculated by two-tailed Welch’s *t*-tests are indicated with stars: * *p* ≤ 0.05, ** *p* ≤ 0.01, *** *p* ≤ 0.001.

**Figure 7 metabolites-12-00306-f007:**
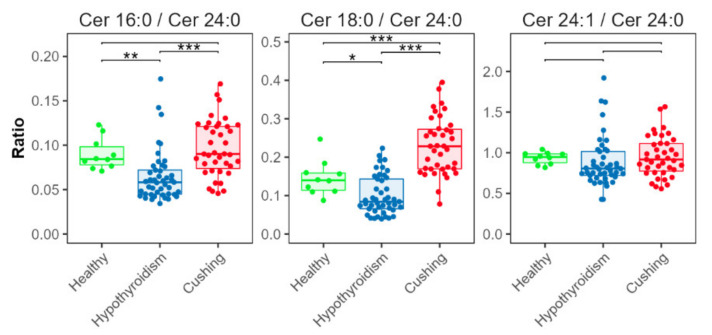
Ceramide ratios. Concentration ratios of Cer d18:1/16:0 (indicated as Cer 16:0), Cer d18:1/18:0 (Cer 18:0), and Cer d18:1/24:1 (Cer 24:1) with Cer d18:1/24:0 (Cer 24:0). Levels of statistical significance between two groups calculated by two-tailed Welch’s *t*-tests are indicated with stars: * *p* ≤ 0.05, ** *p* ≤ 0.01, *** *p* ≤ 0.001.

**Table 1 metabolites-12-00306-t001:** General characteristics (median and range) of healthy beagle (HB) dogs, dogs with hypothyroidism (HT), and dogs with Cushing’s syndrome (CS).

Characteristics	Healthy Beagle (HB)	Dogs with HT	Dogs with CS
Age (years)	6 (2–7)	6 (3–14)	11 (5–16) ^a,b^
Body weight (kg)	14.8 (10–18.9)	nk	14.3 (2.1–42.9)
Females (n)	3	23 ^c,d^	18 (15 spayed)
Males (n)	7	18 ^c,d^	21 (13 castrated)

nk: not known, ^a^ significant difference compared with HB (*p* < 0.0001), ^b^ significant difference compared with HT (*p* < 0.0001), ^c^ neuter status unknown, ^d^ gender of four dogs with HT unknown.

**Table 2 metabolites-12-00306-t002:** Parameters (median and range) used to diagnose hypothyroidism (HT) or Cushing’s syndrome (CS).

Measured Parameters	Dogs with HT	Dogs with CS
Total T4 (µg/dL)	<0.7	
TSH (mU/L)	109 (88–316)	
Baseline cortisol (µg/dL)		3.8 (1.3–13.4)
Post-ACTH cortisol (µg/dL)		19.9 (8.3–47.3)
0 h cortisol before dexamethasone injection (µg/dL)		4.3 (1.5–13.6)
4 h cortisol after dexamethasone injection (µg/dL)		1.05 (0.2–8.8)
8 h cortisol after dexamethasone injection (µg/dL)		1.9 (0.5–6.9)

## Data Availability

The lipidomics dataset including the mass spectrometry raw data and R scripts to process the data and plot figures in this manuscript are available from the authors. The data are not publicly available due to ethical restrictions.

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
