# Peer review of "Serum Lipidome Signatures of Dogs with Different Endocrinopathies Associated with Hyperlipidemia"

_metabolites, 2022, doi:10.3390/metabo12040306_

Round 1

Reviewer 1 Report

The overall presentation of data is clear and seems sound.

1. It is suggested that the basic data on the diagnoses of hypothyroidism and Cushing's syndrome be presented in the main part of the MS rather than in an appendix eg fT4 and TSH for hypothyroidism, and post-dexamethasone serum cortisol and ACTH for Cushing's.

2. The current table 1 legend refers to TSH but no TSH results are shown.

3. TSH measurement is stated to be in ng/mL but this is not standard and conversion to mU/L would be preferable.

4. Do the findings have any impact on clinical veterinary practice?

Author Response

Answers to Reviewer’s suggestions

The authors thank the editor and the reviewers for their invested time and the important suggestions.

Please find a point-by-point response to every reviewer’s comments below. The newly corrected parts are marked with the “Track Change System” within the manuscript.

We hope that these changes are sufficient to result in acceptance of the manuscript.

Reviewer 1

The overall presentation of data is clear and seems sound.

We thank the reviewer for this nice remark.

  1. It is suggested that the basic data on the diagnoses of hypothyroidism and Cushing's syndrome be presented in the main part of the MS rather than in an appendix eg fT4 and TSH for hypothyroidism, and post-dexamethasone serum cortisol and ACTH for Cushing's.

We thank the reviewer for this remark. All the basic data on the diagnosis of hypothyroidism and Cushing’s syndrome is now presented in table 2 included in the main manuscript (material and method section, lines 375-378).

  1. The current table 1 legend refers to TSH but no TSH results are shown.

We thank the reviewer for spotting this and apologize for the mistake. TSH has been deleted from the table 1 legend.

  1. TSH measurement is stated to be in ng/mL but this is not standard and conversion to mU/L would be preferable.

The TSH values have been converted to mU/L (table 2 and material and method section lines 331 and 338).

  1. Do the findings have any impact on clinical veterinary practice?

We thank the reviewer for this remark. In the authors opinion the finding won’t not yet have an impact on clinical veterinary practice; however, the results can be starting points for further interesting research. For example, seem the ceramides and their ratios very interesting. In human medicine they are associated with many pathological states and are begun to be used in the routine laboratory. Further research on ceramide concentrations in dogs with diseases related to hyperlipidemia (e.g. pancreatitis, hepatobiliary disease, diabetes mellitus) would therefore be extremely interesting and possibly will lead to a clinical impact in dogs in future.

Reviewer 2 Report

The paper of Sieber-Ruckstuhl and colleagues is aimed to evaluate the consequences of Cushing Syndrome and Hypothyroidism on lipid profile in dogs.

The aim of the study is intriguing. The methods is adequate. The results are clearly presented. The discussion is coherent with the results.

Only minor points should be clarified:

  • It is not stated if the dogs enrolled were already affected by cardiovascular diseases. It is reported only the exclusion of patients with chronic kidney disease or diabetes mellitus. This is a relevant aspect in order to interpretate the results.
  • In the discussion the authors compared their results with those observed in humans. However, it should be better discussed how their results could be useful in human lipid characterization.

Author Response

Answers to Reviewer’s suggestions

The authors thank the editor and the reviewers for their invested time and the important suggestions.

Please find a point-by-point response to every reviewer’s comments below. The newly corrected parts are marked with the “Track Change System” within the manuscript.

We hope that these changes are sufficient to result in acceptance of the manuscript.

Reviewer 2

The paper of Sieber-Ruckstuhl and colleagues is aimed to evaluate the consequences of Cushing Syndrome and Hypothyroidism on lipid profile in dogs.

The aim of the study is intriguing. The methods is adequate. The results are clearly presented. The discussion is coherent with the results.

We thank the reviewer for these nice remarks. 

Only minor points should be clarified:

  • It is not stated if the dogs enrolled were already affected by cardiovascular diseases. It is reported only the exclusion of patients with chronic kidney disease or diabetes mellitus. This is a relevant aspect in order to interpretate the results.

We thank the reviewer for this very important point. In veterinary practice the cardiovascular system is not routinely evaluated. Usually, dogs only get a thorough clinical examination including heart auscultation. The vascular system is not routinely assessed, unless there is a specific suspicion (e.g. thromboembolism), where then a general anesthesia with contrast CT can be performed. We added some sentences in the limitation section to clarify this (lines 315-320).  

  • In the discussion the authors compared their results with those observed in humans. However, it should be better discussed how their results could be useful in human lipid characterization.

In the authors opinion the results in the ceramides and their ratios and the sphingolipids seem intriguing. In human medicine these lipids are already associated with many pathological states and ceramides are increasingly being used in the routine laboratory to assess the risk for cardiovascular disease. The results herein seem to confirm a connection of increased ceramides and cardiovascular disease and changes in the sphingolipids and cardiovascular health. Ceramides and sphingolipids seem therefore a very interesting topic for future cardiovascular research, in humans and other animals. The authors tried to emphasize these two points within the discussion section (lines 274, 294-295).
